# END-TO-END HIERARCHICAL TEXT CLASSIFICATION WITH LABEL ASSIGNMENT POLICY

## ABSTRACT

We present an end-to-end reinforcement learning approach to hierarchical text classification where documents are labeled by placing them at the right positions in a given hierarchy. While existing "global" methods construct hierarchical losses for model training, they either make "local" decisions at each hierarchy node or ignore the hierarchy structure during inference. To close the gap between training/inference and capture label dependencies in an end-to-end manner, we propose to learn a **l**abel **a**ssignment **p**olicy to determine *where* to place the documents and *when* to stop. The proposed method, HiLAP, directly optimizes metrics over the hierarchy, makes *inter-dependent* decisions during inference, and can be combined with different text encoding models for end-to-end training. Experiments on three public datasets show that HiLAP yields an average improvement of 33.4% in Macro-F1 and 5.0% in Samples-F1, outperforming state-of-the-art methods by a large margin.[1]

## 1 INTRODUCTION

In recent years there has been a surge of interest in leveraging taxonomies and hierarchies to organize and classify text documents, leading to the development of hierarchical text classification (HTC) methods—methods that can predict for a document multiple appropriate labels (which together constitute a sub-tree) in a given hierarchy. These methods have found a wide range of applications such as question answering (Qu et al., 2012), online advertising (Agrawal et al., 2013), and scientific literature organization (Peng et al., 2016). In contrast to traditional "flat" classification, the key challenge of HTC lies in modeling the inter-dependent, large-scale, and imbalanced label space.

Due to the complexity of HTC, how to better utilize the label hierarchy remains an open problem. HTC methods are traditionally divided into three categories, namely *flat*, *local*, and *global* approaches (Silla & Freitas, 2011). Flat approaches generally ignore the label hierarchy. Some only predict labels at the leaf nodes and then add all the ancestors of the predicted leaf nodes. Others ignore the hierarchy and perform standard multi-label classification, in which inconsistencies (*i.e.*, one label is predicted but its ancestors are not) may occur and post-processing is thus needed. Local approaches train a set of local classifiers per node/per parent node/per level, which function independently and (usually) make predictions in a top-down order to avoid inconsistencies. Traditional global approaches (Cai & Hofmann, 2004; Vens et al., 2008; Silla Jr & Freitas, 2009) are largely modified based on specific flat models and rely on static, human curated features as input. In addition, many existing global approaches make unrealistic assumptions of the problem as in flat approaches. For example, Hierarchical-SVM (Cai & Hofmann, 2004) requires that all possible labels are on the leaf nodes and the heights of leaf nodes are the same.

Recent approaches (Kim, 2014; Lai et al., 2015; Yang et al., 2016) to text classification mainly focus on flat classification and have been shown to be very effective. However, their performance in HTC is relatively less studied. Even if the classification task is essentially hierarchical, prior work (Gopal & Yang, 2013; Johnson & Zhang, 2014; Peng et al., 2016; 2018) still makes *flat* and *independent* predictions and utilizes intuitive constraints, such as the embeddings of one label and its parent should be close. One recent framework (Wehrmann et al., 2018) attempted to leverage both local and global information. However, it uses static features and its inference is essentially flat, which may lead to inconsistencies.

---

[1]Code and data will be released on GitHub upon acceptance.

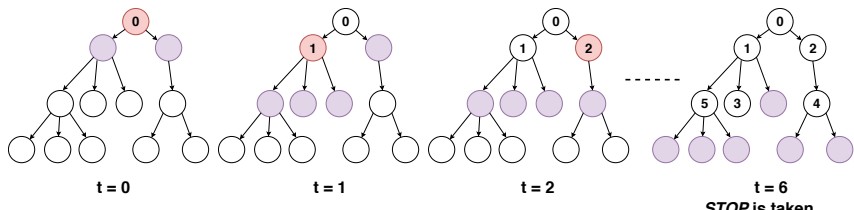

Figure 1: **An illustrative example showing our label assignment policy.** At $t = 0$, the document $x_i$ is placed at the root label and the policy would decide if $x_i$ should be placed to its two adjacent (denoted by *purple*) labels. At $t = 1$, $x_i$ is placed at label 1, which adds another three adjacent labels as the candidates. At $t = 6$, the *stop* action is taken and the label assignment process is thus terminated. We then take all the labels where $x_i$ has been placed (a sub-tree consisting of label 0, 1, ..., 5) as $x_i$'s document labels.

In this paper, we present an end-to-end reinforcement learning approach to HTC where documents are labeled by placing them at the right positions in a label hierarchy. We propose HiLAP, a principled global framework that learns a label assignment policy to determine *where* to place the documents and *when* to stop. HiLAP optimizes its policy by exploring the label hierarchy, in which training and inference follow the same routine and *inter-dependent* decisions are made. Compared to flat and local approaches, HiLAP achieves better effectiveness because it examines the global hierarchical structure during *both training and inference* phases. Compared to most existing global approaches, HiLAP has more flexibility in that it has no constraints on the structure of the hierarchy. Furthermore, the label assignment policy of HiLAP ensures that its predictions are always *consistent* and no post-processing is needed.

HiLAP can be combined with different text representation learning models and trained in an end-to-end fashion. We select three representative text encoding models as the *base models* to evaluate the effectiveness of HiLAP. Experiments on three public datasets from different domains show that combining HiLAP with existing representation learning models yields an average performance improvement of 33.4% in Macro-F1 and 5.0% in Samples-F1, outperforming state-of-the-art HTC methods by a large margin. In particular, ablation study shows that HiLAP is especially beneficial to those unpopular labels at the bottom levels.

## 2 LEARNING LABEL ASSIGNMENT POLICY FOR END-TO-END HIERARCHICAL TEXT CLASSIFICATION

This section presents the proposed end-to-end reinforcement learning approach to hierarchical text classification. We first introduce our label assignment policy including the design of its actions and rewards, and then describe the details of policy learning.

### 2.1 HIERARCHICAL LABEL ASSIGNMENT

We define a label hierarchy $\mathbf{H} = (\mathbf{L}, \mathbf{E})$ as a tree or DAG (directed acyclic graph)-structured hierarchy with node set $\mathbf{L}$ (*i.e.*, the labels), and edge set $\mathbf{E}$ (which indicates the parent-child relationship between labels). Taking a set of documents $\{x_1, x_2, ..., x_N\}$ and their document labels $\{L_1, L_2, ..., L_N\} \in \mathbf{L}$ as input, we aim to learn a policy $\mathcal{P}$ to place each documents $x_i$ to its labels $L_i$ on the label hierarchy $\mathbf{H}$. Specifically, the policy $\mathcal{P}$ puts $x_i$ at the root label in the beginning and at each time step, decides which label $x_i$ should be further placed to, among all the *adjacent* labels of where $x_i$ has been placed, until a special *stop* action is taken. An illustration of our label assignment policy is shown in Figure 1. We define one *base model* $\mathcal{B}$ as a mapping $f$ that converts each raw document $x_i$ to a finite dimensional vector as its representation, *i.e.*, the document embedding $\mathbf{e}_d \in \mathbb{R}^D$ ($D$ denotes the embedding size). $\mathcal{B}$ can be any neural text representation learning model and its output $\mathbf{e}_d$ is used as the input of the policy $\mathcal{P}$. The challenge, compared to standard classification setup, is that we need to model $\mathbf{E}$, *i.e.*, the relationship between labels.

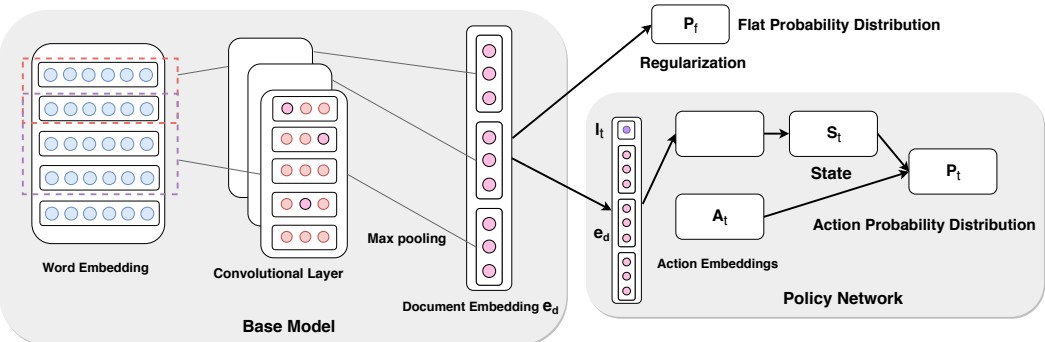

Figure 2: **The architecture of the proposed framework HiLAP**. One CNN model (Kim, 2014) is used as the base model for illustration. The document embedding $\mathbf{e}_d$ generated by the base model is combined with the label embedding $\mathbf{l}_t$ and used as the state representation $\mathbf{s}_t$, based on which actions are taken by the policy network.

## 2.2 REINFORCEMENT LEARNING FOR LABEL ASSIGNMENT

To learn the label assignment policy, we train a policy network to determine *where* to place the documents and *when* to stop as follows.

**Actions** We regard the process of placing a document $x_i$ to the right positions on the label hierarchy as making a sequence of actions. Specifically, we define that an action $a_t$ at time step $t$ is to select one label $l_t$ from the action space $A_t$ and place $x_i$ to that label $l_t$. We denote the children of label $l_t$ as $\mathcal{C}(l_t)$. The action space $A_t$ consists of all the adjacent labels of where the document $x_i$ has been placed. In this way, for example, HiLAP can first place $x_i$ to a label at level 3 if the confidence (probability) of that label is higher and then place it to another label at level 1.

At the beginning of each episode, $x_i$ is placed at the root label $l_0$ and the action space $A_t = \mathcal{C}(l_0)$, *i.e.*, all the labels at level 1. When $x_i$ is placed at another label $l_1$, its children $\mathcal{C}(l_1)$ would then be added to the action space $A_t$. In addition, one *stop* action $\mathbf{e}_{\text{stop}} \in \mathbb{R}^C$ ($C$ denotes the embedding size) is added to the action space $A_t$ so that the model can learn when to stop placing document $x_i$ to new labels. Intuitively, when the confidence of placing $x_i$ to another label is lower than the *stop* action, the label assignment process would be terminated. In short, the size of $A_t$ is $|\{\mathcal{C}(l_0), \mathcal{C}(l_1), ..., \mathcal{C}(l_t), stop\} - \{l_1, l_2, ..., l_t\}|$. Note that in other local/global approaches the predictions on different paths are *independent* while in HiLAP the inter-dependencies of labels across different paths and levels are considered and we optimize the metrics over the hierarchy by providing the policy network with rewards that capture the overall quality of its label assignment.

**Rewards** HiLAP receives rewards from the environment as feedback for its actions. Different from existing work where each label of one sample is treated independently, HiLAP captures the quality of all the labels assigned to each sample $x_i$ by rewarding the agent with the *per-sample F1* (see Section 3.1 for details): $\text{F1}^{x_i} = \frac{2 \times P^{x_i} \times R^{x_i}}{P^{x_i} + R^{x_i}}$, where $P^{x_i}$ and $R^{x_i}$ denote per-sample precision and recall respectively that compare the predicted labels with gold labels of $x_i$.

Instead of waiting until the end of the label assignment process and comparing the predicted labels with the gold labels, we use reward shaping (Mao et al., 2018), *i.e.*, giving intermediate rewards at each time step, to accelerate the learning process. Specifically, we set the reward $r$ of $x_i$ at time step $t$ to be the difference of its F1 score between current and last time step: $r_t^{x_i} = \text{F1}_t^{x_i} - \text{F1}_{t-1}^{x_i}$. If current F1 is better than that at last time step, the reward would be positive, and vice versa. The cumulative reward from current time step to the end of an episode would cancel the intermediate rewards and thus reflect whether current action improves the overall label assignment of one sample or not. As a result, the learned policy would not focus on the current placement but have a long-term view that takes following actions into account.

**Policy Network** We parameterize each action $a_t$ by a policy network $\pi(\mathbf{a} \mid \mathbf{s}; \mathbf{W})$. The architecture of HiLAP is shown in Figure 2. For each document, its representation $\mathbf{e}_d$ is generated by the base model $\mathcal{B}$. For each label, a label embedding $\mathbf{l} \in \mathbb{R}^C$ is randomly initialized and updated during training. To model the label relationship $\mathbf{E}$, the embeddings of the document $\mathbf{e}_d$ and current label $\mathbf{l}_t$

are concatenated and projected to a vector $\mathbf{s}_t \in \mathbb{R}^C$ via a two-layer feed-forward network. $\mathbf{s}_t$ has the same size as the label embedding $\mathbf{l}$ and we use $\mathbf{s}_t$ as the state representation of the document at current label $\mathbf{l}_t$. By stacking the action embeddings, we can obtain an action matrix $\mathbf{A}_t$ with size $|\{\mathcal{C}(l_0), \mathcal{C}(l_1), ..., \mathcal{C}(l_t), stop\} - \{l_1, l_2, ..., l_t\}| \times C$. $\mathbf{A}_t$ is multiplied with the state embedding $\mathbf{s}_t$, which outputs the probability distribution of actions. Finally, an action $a_t$ is sampled based on the probability distribution of the action space:

$$\mathbf{s}_t = \text{ReLU}(\mathbf{W}_l^1 \text{ReLU}(\mathbf{W}_l^2[\mathbf{e}_d; \mathbf{l}_t])),$$
$$\pi(\mathbf{a} \mid \mathbf{s}; \mathbf{W}) = \text{softmax}(\mathbf{A}_t \mathbf{s}_t),$$
$$a_t \sim \pi(\mathbf{a} \mid \mathbf{s}; \mathbf{W}).$$

We use REINFORCE (Williams, 1992), one instance of the policy gradient methods, as the optimization algorithm. To reduce variance, 10 rollouts for each training sample are run and the rewards are averaged. In addition, we adopt a *self-critical* training approach (Rennie et al., 2017). For each document $x_i$, two label assignments are generated: $\tilde{L}_{x_i}$ is sampled from the probability distribution, and $\hat{L}_{x_i}$, the baseline label assignment, is greedily generated by choosing the action with the highest probability at each time step. We use $r(\tilde{L}_{x_i}) - r(\hat{L}_{x_i})$ as the actual reward, which ensures that the policy network learns to place the document to positions with higher F1 score than the greedy baseline. At the time of inference, we greedily select labels with the highest probability as $\hat{L}_{x_i}$.

### 2.3 TOP-DOWN SUPERVISED PRE-TRAINING

It is known that reinforcement learning models often suffer from high variance during training. Instead of learning from scratch, we use supervised learning to pre-train our framework. We denote the supervised variant as HiLAP-SL. While most parameters of HiLAP-SL are shared with HiLAP, its action space and way of exploring of the label hierarchy $\mathbf{H}$ are dissimilar.

The main difference is that HiLAP-SL explores the label hierarchy $\mathbf{H}$ in a top-down manner. At each time step $t$, the document goes down one level on the hierarchy. HiLAP-SL concentrates on the local discrimination of labels with the same parent. The local per-parent label probability distribution $\mathbf{p}_t^{\text{SL}}$ is generated as follows.

$$\mathbf{p}_t^{\text{SL}} = \sigma(\mathbf{C}_t \mathbf{s}_t),$$

where $\sigma$ denotes the sigmoid function, and $\mathbf{C}_t \in \mathbb{R}^{|\mathcal{C}(l_t)| \times C}$ denotes the action space of HiLAP-SL, *i.e.*, an embedding matrix consisting of the children of current label $l_t$ (rather than all the labels where $x_i$ has been placed as in HiLAP).

Another difference is that in HiLAP the actions are sampled and thus the documents might be placed to wrong labels, while in HiLAP-SL only the ground-truth positions are traversed during training. Specifically, if there are $K(\geq 1)$ *ground-truth* labels at the same level, the document embedding $\mathbf{e}_d$ would be cloned $K$ times following each label and $K$ different paths would be generated *independently*. The loss function of HiLAP-SL is defined as follows.

$$\mathcal{L}_l = \sum_{t=0}^{T} \mathcal{L}_t,$$

where $T$ is the lowest label's height of one sample ($T$ may vary by samples) and $\mathcal{L}_t$ estimates the binary cross entropy over the candidate labels $\mathcal{C}(l_t)$ at each time step $t$. HiLAP-SL works as if there were a set of local classifiers, although most of its parameters (except for the label embeddings $\mathbf{l}_t$) are shared by all the labels so that one does not need to actually train a set of classifiers. During inference, HiLAP-SL follows the same top-down routine as in training using the per-parent label probability $\mathbf{p}_t^{\text{SL}}$ and thus no post-processing is needed for inconsistency correction.

All the parameters of HiLAP are shared with HiLAP-SL and can be initialized by the pre-trained HiLAP-SL model except for the embedding of the *stop* action $\mathbf{e}_{\text{stop}}$ (which is randomly initialized).

### 2.4 COMBINING FLAT, LOCAL, AND GLOBAL INFORMATION FOR POLICY LEARNING

We further add a *flat component* to our framework as a regularization of the base model. Specifically, the flat component is a simple feed-forward network consisting of a fully connected layer and the

sigmoid function. It projects the document embedding $\mathbf{e}_d$ to a label probability distribution $\mathbf{p}_f$ of all the labels $\mathbf{L}$ on the hierarchy.

$$\mathbf{p}_f = \sigma(\mathbf{W}_f \mathbf{e}_d).$$

The combination of the base model and the flat component is exactly the same as a flat model and ensures that the document representation $\mathbf{e}_d$ learned by the base model $\mathcal{B}$ has the capability of flat classification among all the labels $\mathbf{L}$. We use a flat loss $\mathcal{L}_f$ to measure the binary cross entropy over all the labels as in the flat models. Combining the flat loss with local loss, the supervised loss in HiLAP-SL is defined as $\mathcal{L}_{SL} = \lambda \mathcal{L}_f + (1 - \lambda) \sum_{t=0}^{T} \mathcal{L}_t$, where $\lambda \in [0, 1]$ is the mixed ratio. Similar to Celikyilmaz et al. (2018), we also found that mixing a proportion of the supervised loss is beneficial to the learning process of HiLAP. Further combining the global information, the total loss of HiLAP is defined as $\mathcal{L}_{mixed} = \mathcal{L}_{RL} + \alpha \mathcal{L}_{SL}$, where $\alpha$ is a scaling factor accounting for the difference in magnitude between $\mathcal{L}_{RL}$ and $\mathcal{L}_{SL}$. While we do not directly use the flat component during inference, it helps the learning process of the base model and improves the performance of both HiLAP-SL and HiLAP, which we will show in Section 3.5.

## 3 Experiments

We evaluate the benefits of our framework against a number of state-of-the-art HTC approaches, with the goal of answering the following questions:

**Q1** How does our proposed method (HiLAP) compare to state-of-the-art HTC approaches?

**Q2** How does HiLAP compare to other hierarchical classification frameworks when the same base models are adopted by all the frameworks?

**Q3** How do different components in HiLAP contribute to its performance in terms of the popular(sparse) labels?

### 3.1 Datasets and Evaluation Metrics

We conduct experiments on three public and commonly used datasets from different domains. The first two datasets are related to news categorization, including RCV1 (Lewis et al., 2004) and the New York Times (NYT) annotated corpus (Sandhaus, 2008). We follow the original training/test split for RCV1 and sub-sample NYT due to its large size. The third dataset is the Yelp Dataset Challenge 2018[2], which consists of a subset of Yelp businesses and their reviews. We use the Yelp Business Categories[3] as the label hierarchy and predict the categories of one business using its reviews. For each dataset, there may be more than one label at each level and the lowest labels of a sample may not be at the leaf nodes. A summary of the datasets is shown in Table 1 and further details can be found in Appendix A.

Table 1: Summary of the three datasets. $|\mathbf{L}|$ denotes the number of labels in the label hierarchy. $\text{Avg}(|L_i|)$ and $\text{Max}(|L_i|)$ denote the average and maximum number of labels of one sample, respectively.

| Dataset | Taxonomy | $|\mathbf{L}|$ | $\text{Avg}(|L_i|)$ | $\text{Max}(|L_i|)$ | Training | Test |
|---------|----------|------|-----------|-----------|----------|---------|
| RCV1 | Tree | 103 | 3.24 | 17 | 23,149 | 781,265 |
| NYT | Tree | 115 | 2.52 | 14 | 25,279 | 10,828 |
| Yelp | DAG | 539 | 3.77 | 32 | 87,375 | 37,265 |

We use standard metrics (Johnson & Zhang, 2014; Peng et al., 2016) for HTC including Micro-F1, Macro-F1, and Samples-F1[4]. Micro-F1 measures the overall precision/recall and favors labels with more samples. Macro-F1 calculates the F1 scores of all the labels and performs an unweighted average over them. Similarly, Samples-F1 calculates the F1 scores of all the samples and averages them ($\frac{\sum_i \text{F1}^{x_i}}{N}$). Recall that $\text{F1}^{x_i}$ is used as the reward in HiLAP.

---

[2]https://www.yelp.com/dataset/challenge

[3]https://www.yelp.com/developers/documentation/v3/all_category_list

[4]We take the name *Samples-F1* from *sklearn.metrics.f1_score*. It is referred to as *Example-based F1* in Partalas et al. (2015), *EBF* in Peng et al. (2016) and *Edge-F1* in Mao et al. (2018).

## 3.2 BASE MODELS

In our experiments, three representative text classification models with different characteristics are selected as the base models to prove the robustness and versatility of HiLAP. As one will see, HiLAP consistently improves the base model through better exploration of the label hierarchies.

**TextCNN** (Kim, 2014) is the classic convolutional neural network for text classification. In our implementation, TextCNN is composed of one convolutional layer with three kernels of different sizes (3, 4, 5), followed by max pooling, a dropout layer, and a fully-connected layer. We chose TextCNN because it was one of the first successful and well used neural-based models for text classification.

**HAN** (Yang et al., 2016) first learns the representation of sentences by feeding words in each sentence to a GRU-based sequence encoder (Bahdanau et al., 2014) and then feeds the representation of the encoded sentences into another GRU-based sequence encoder, which generates the representation of the whole document. Attention mechanism such as word attention and sentence attention is also used. We chose HAN because it uses RNNs instead of CNNs and is shown to be effective on the Yelp Review datasets (Zhang et al., 2015).

**bow-CNN** (Johnson & Zhang, 2014) employs bag of words (multi-hot zero-one vectors) as input and directly applies CNN to high-dimensional text data. It learns the representation of small text regions (rather than single words) for use in classification. We chose bow-CNN since it does not use any word embeddings as in TextCNN and HAN. In addition, bow-CNN achieved the state-of-the-art performance on the RCV1 dataset (Lewis et al., 2004).

## 3.3 BASELINES

We compare our framework with state-of-the-art HTC methods. The traditional methods that we compare with are Support Vector Machines (SVM) and its hierarchical variants. Specifically, SVM performs standard multi-label classification using one-vs-the-rest (OvR) strategy. Leaf-SVM treats each leaf node as a label and adds the ancestors of predicted leaf nodes. Other variants include HSVM (Tsochantaridis et al., 2005), Top-Down SVM (TD-SVM) (Liu et al., 2005), and Hierarchically Regularized Support Vector Machines (HR-SVM) (Gopal & Yang, 2013). The neural-based methods that we compare with include HLSTM (Chen et al., 2016), HR-DGCNN (Peng et al., 2018) and HMCN (Wehrmann et al., 2018). We also compare with the base models, namely, TextCNN (Kim, 2014), HAN (Yang et al., 2016), and bow-CNN (Johnson & Zhang, 2014) to see how much we could improve upon them via better exploration of the hierarchy.

## 3.4 IMPLEMENTATION DETAILS

For each base model, we follow its original implementation and fix the hyper-parameters for different datasets. We randomly sample 10% from the training set as development set (Johnson & Zhang, 2014; Peng et al., 2018). We set batch size to 32 and use the first 256 tokens of each document for representation learning. We use a constant threshold (0.5) for all the labels. All the models are trained using an Adam optimizer with initial learning rate 1e-3 and weight decay 1e-6. We use pre-trained GloVe word vectors (Pennington et al., 2014) with dimensionality 50 as word embeddings for TextCNN and HAN. We limit the vocabulary to the most frequent 30000 words in the training data and generate multi-hot vectors as the input of bow-CNN. For our framework, we set the size of $\mathbf{W}_l^2$ to 500 and the sizes of $\mathbf{W}_l^1$ and label embedding $\mathbf{l}_t$ to 50.

## 3.5 EXPERIMENTAL RESULTS

**Main Results** Table 2 and 3 compare the performance of HiLAP to the state-of-the-art HTC baselines. These results provide positive answers to our question **Q1**: On the RCV1 dataset, HiLAP (HAN) achieves similar performance to HR-DGCNN even though the original flat HAN is worse than HR-DGCNN. Our HiLAP (TextCNN) outperforms most baselines in Macro-F1 and HiLAP

(bow-CNN) achieves the best performance on all the three metrics.[5] On the NYT dataset, similar results are observed: TextCNN and HAN are both improved when combining with HiLAP and HiLAP (bow-CNN) again achieves the best performance. On the Yelp dataset, HiLAP (HAN) achieves the best Micro-F1 and Samples-F1, while HiLAP (bow-CNN) obtains the highest Macro-F1. Interestingly, a simple SVM outperforms several neural-based models, indicating that traditional feature-based methods still play an important role in HTC.

Table 2: Comparison results on the RCV1 dataset. * denotes the results reported in Peng et al. (2018) on the same dataset split. Note that the original results of RCV1 in Gopal & Yang (2013) are not comparable because they used a different label hierarchy.

| | Method | Micro-F1 | Macro-F1 | Samples-F1 |
|---|---|---|---|---|
| Flat | Leaf-SVM* | 69.1 | 33.0 | - |
| | SVM | 80.4 | 46.2 | 80.5 |
| | HLSTM* (Chen et al., 2016) | 67.3 | 31.0 | - |
| | TextCNN (Kim, 2014) | 76.6 | 43.0 | 75.8 |
| | HAN (Yang et al., 2016) | 75.3 | 40.6 | 76.1 |
| | bow-CNN (Johnson & Zhang, 2014) | 82.7 | 44.7 | 83.3 |
| Local & Global | TD-SVM (Liu et al., 2005) | 80.1 | 50.7 | 80.5 |
| | HSVM* (Tsochantaridis et al., 2005) | 69.3 | 33.3 | - |
| | HR-SVM* (Gopal & Yang, 2013) | 72.8 | 38.6 | - |
| | HR-DGCNN* (Peng et al., 2018) | 76.1 | 43.2 | - |
| | HMCN (Wehrmann et al., 2018) | 80.8 | 54.6 | 82.2 |
| | **HiLAP** (TextCNN) | 78.6 | 50.5 | 80.1 |
| | **HiLAP** (HAN) | 75.4 | 45.5 | 77.4 |
| | **HiLAP** (bow-CNN) | **83.3** | **60.1** | **85.0** |

Table 3: Results of various methods on the NYT and Yelp datasets.

| | Method | NYT | | | Yelp | | |
|---|---|---|---|---|---|---|---|
| | | Micro-F1 | Macro-F1 | Samples-F1 | Micro-F1 | Macro-F1 | Samples-F1 |
| Flat | SVM | 72.4 | 37.1 | 74.0 | 66.9 | 36.3 | 68.0 |
| | TextCNN (Kim, 2014) | 69.5 | 39.5 | 71.6 | 62.8 | 27.3 | 63.1 |
| | HAN (Yang et al., 2016) | 62.8 | 22.8 | 65.5 | 66.7 | 29.0 | 67.9 |
| | bow-CNN (Johnson & Zhang, 2014) | 72.9 | 33.4 | 74.1 | 63.6 | 23.9 | 63.9 |
| Local & Global | TD-SVM (Liu et al., 2005) | 73.7 | 43.7 | 75.0 | 67.2 | 40.5 | 67.8 |
| | HMCN (Wehrmann et al., 2018) | 72.2 | 47.4 | 74.2 | 66.4 | 42.7 | 67.6 |
| | **HiLAP** (TextCNN) | 69.9 | 43.2 | 72.8 | 65.5 | 37.3 | 68.4 |
| | **HiLAP** (HAN) | 65.2 | 28.7 | 68.0 | **69.7** | 38.1 | **72.4** |
| | **HiLAP** (bow-CNN) | **74.6** | **51.6** | **76.6** | 68.9 | **42.8** | 71.5 |

**Performance Comparison using the Same Text Encoding Model** To answer **Q2** and **Q3**, we compare different frameworks that support the use of *exactly the same* base model and Figure 3 summarizes the comparison results.[6] As one may notice, due to the extreme imbalance of the datasets, directly applying a flat multi-label classification model may suffer from low Macro-F1, *i.e.*, the predictions of flat models are inevitably biased to the most popular labels. HMCN also has the same issue, resulting in Macro-F1 scores lower than 10 when combining with some base models. In contrast, HiLAP significantly outperforms the baselines especially in Macro-F1, which implies that our policy network is better at tackling labels with relatively few samples. On the downside, it is also observed that HiLAP-SL may have a negative effect in terms of Micro-F1 (although it is usually marginal compared with the gain in Macro-F1). However, such negative effects are eliminated by HiLAP through better exploration of the label hierarchy **H**. Overall, HiLAP obtains the highest performance on 24/27 results across three datasets, three base models, and three evaluation metrics.

**Ablation Study of Framework Components in HiLAP** To better understand **Q3**, We show the ablation analysis of our framework in Table 4. Using *Flat Component Only* degenerates our framework to the flat baseline. By comparing the results of *Flat Component Only* and HiLAP-SL-NoFlat (a variant of HiLAP-SL without flat loss), we further confirm that flat approaches are likely to neglect sparse labels, which results in low Macro-F1. By combining the two components, HiLAP-SL

---

[5]Our results are not directly comparable with Johnson & Zhang (2014) and Lewis et al. (2004) due to implementation details and the fact that they tune the threshold for each label using k-fold cross-validation. See Appendix B for more discussions of the baselines.

[6]For HMCN, we replace its static features with a base model for fair comparison. The original HMCN + HAN failed to learn and we removed its batch normalization. See Appendix B for more details.

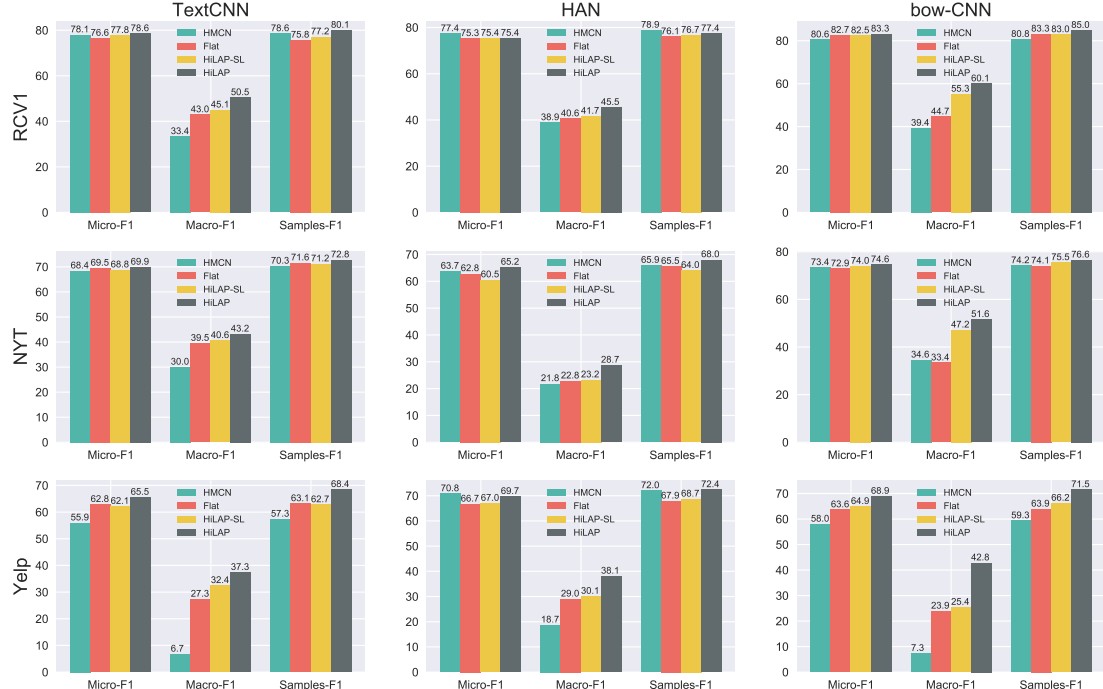

Figure 3: Performance comparison of different frameworks using the same base model as input. We improved HMCN + HAN by removing its batch normalization.

Table 4: Ablation study of HiLAP-SL and HiLAP when combining with bow-CNN (Johnson & Zhang, 2014) on the RCV1 dataset (Lewis et al., 2004).

| Method | Micro-F1 | Macro-F1 | Samples-F1 |
|---|---|---|---|
| Flat Component Only | 82.7 | 44.7 | 83.3 |
| HiLAP-SL-NoFlat | 81.0 | 52.1 | 81.7 |
| HiLAP-SL | 82.5 | 55.3 | 83.0 |
| HiLAP-NoSL | 83.2 | 59.3 | 85.0 |
| HiLAP-NoFlat | 83.0 | 59.8 | 84.7 |
| HiLAP | **83.3** | **60.1** | **85.0** |

achieves close performance to *Flat Component Only* on Micro-F1 and Samples-F1 and even higher Macro-F1 than HiLAP-SL-NoFlat. HiLAP-NoSL is initialized by the pre-trained HiLAP-SL model without mixing the supervised loss during its training. We can see that using the reinforced loss alone still improves the performance on all the three metrics. After removing the flat loss during the training of HiLAP, HiLAP-NoFlat shows slightly lower performance than the full HiLAP model, indicating that the flat component serves as a regularization of the base model and is beneficial to the overall performance.

We also analyze the source of the performance gains by dividing the labels based on their levels and number of supporting samples. Figure 4 shows one comparison of absolute Macro-F1 differences between various methods and the base model. We observe similar trends for other setups/metrics and omit them for better view. As depicted in Figure 4, HiLAP and HiLAP-SL are especially beneficial to those unpopular labels at the bottom levels.

## 4 RELATED WORK

Hierarchical text classification and general hierarchical classification approaches have been developed for many applications. In the biomedical domain, medical subject headings (MeSH) indexing, which is to assign a set of MeSH main headings to citations, has been studied for years (Liu et al., 2015; Peng et al., 2016). In addition, there are plenty of methods focusing on the hierarchical prediction of protein and gene functions (Clare & King, 2003; Silla Jr & Freitas, 2009; Secker et al., 2010;

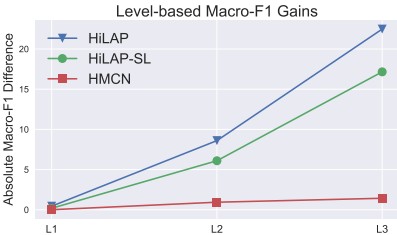 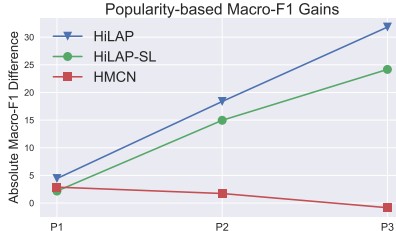

Figure 4: Level-based and popularity-based Macro-F1 gains compared to bow-CNN (Johnson & Zhang, 2014) on the NYT (Sandhaus, 2008) dataset. We show the per-level gains on the left, in which L1, L2, and L3 denote the levels of the hierarchy. We divide the labels into three equal sized categories, namely P1, P2, and P3, in a descending order based on their number of samples, and show their gains on the right.

Cerri et al., 2016). Another line of work concentrates on document categorization. Both traditional methods (Lewis et al., 2004; Gopal & Yang, 2013) and neural methods (Johnson & Zhang, 2014; Peng et al., 2018) have been proposed to classify the topics of newswire and web content (Dumais & Chen, 2000; Sun & Lim, 2001), categories of laws and patents (Bi & Kwok, 2015; Cai & Hofmann, 2004; Rousu et al., 2005).

Many previous works (Liu et al., 2005; Xue et al., 2008; Sun & Lim, 2001) train a set of local classifiers and make predictions in a top-down manner. In particular, (Bi & Kwok, 2015) develop Bayes-optimal predictions that minimize the global risks with the trained model but the model is still locally trained. Such local approaches are not popularly used among recent neural-based hierarchical classification models (Johnson & Zhang, 2014; 2016; Peng et al., 2018) since it is infeasible to train many neural classifiers locally.

Global methods, on the other hand, train only one classifier. Although global methods are desirable, they are relatively less studied due to the complexity of the problem. Existing global models are generally modified based on specific flat models. Hierarchical-SVM (Cai & Hofmann, 2004; 2007; Qiu et al., 2009) generalizes Support Vector Machine (SVM) learning based on discriminant functions that are structured in a way that mirrors the label hierarchy. One limitation is that Hierarchical-SVM only supports balanced tree (all possible labels are presumed to be at the same height in their experiments). Hierarchical naive Bayes (Silla Jr & Freitas, 2009) modifies naive Bayes by updating weights of one's ancestors as well whenever one label's weights are updated. There are other global methods that are based on association rules (Wang et al., 2001), C4.5 (Clare & King, 2003), kernel machines (Rousu et al., 2005), and decision tree (Vens et al., 2008). Constraints such as the regularization that enforces the parameters of one node and its parent to be similar (Gopal & Yang, 2013; 2015) are also proposed to leverage the label hierarchy while maintaining scalability. However, their use of hierarchies is somewhat limited.

In addition to the most relevant prior studies mentioned above, dataless classification (Ha-Thuc & Renders, 2011; Song & Roth, 2014; Meng et al., 2018) leverages the labels in the hierarchy as weak supervision and requires almost no training data. Yu et al. (2014); Bhatia et al. (2015) scale to hundreds of thousands of labels but do not assume and leverage the existence of label hierarchies.

## 5 CONCLUSION

We proposed an end-to-end reinforcement learning approach to hierarchical text classification where documents are labeled by placing them at the right positions in the label hierarchy. The proposed framework makes *consistent* and *inter-dependent* predictions, in which any neural-based representation learning model can be used as a base model and a label assignment policy is learned to determine *where* to place the documents and *when* to stop. Experiments on three public datasets of different domains showed that our approach outperforms state-of-the-art hierarchical text classification methods significantly. In the future, we will explore the effectiveness of the proposed framework on other base models and forms of data (*e.g.*, images). We also plan to mix more losses covering other aspects in the objective function and test whether they could further improve the performance of our framework.

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

# A    DATASET DETAILS

In this section, we describe the details of the datasets. The RCV1 dataset (Lewis et al., 2004) is a manually labeled newswire collection of Reuters News from 1996 to 1997. Its news documents are categorized with three aspects: industries, topics, and regions. We use its topic-based label hierarchy for classification as it has been well used in prior work (Gopal & Yang, 2013; Johnson & Zhang, 2014; Peng et al., 2018; Wehrmann et al., 2018). There are 103 categories and four levels in total including all labels except for the root label in the hierarchy.

The NYT annotated corpus (Sandhaus, 2008) is a collection of New York Times news from 1987 to 2007. Due to its large size, we randomly sampled 36107 documents from all the news documents, and further split them into training and test set of 25279 and 10828 samples, respectively. We use the first 3 layers in the hierarchy and keep the labels with at least 40 supporting samples.

For the preprocessing of the Yelp dataset, we first removed categories that have fewer than 100 businesses and then businesses that have fewer than 5 reviews. We concatenated (at most) the first 10 reviews of each business as its representation. We set the training/test ratio to 70%/30%, which results in a training set of 87,375 samples and a test set of 37,517 samples. This is an even more challenging task because the reviews are usually written in an informal way and it is more imbalanced than the RCV1 or NYT datasets (*e.g.*, label *Restaurants* has 32,357 businesses in the training set while *Retirement Homes* has 23).

# B    PERFORMANCE ANALYSIS OF BASELINES

There are several things to note in terms of the performance of the baselines. First, our results are not directly comparable to Lewis et al. (2004); Johnson & Zhang (2014) due to implementation details and the fact that they tune the threshold for each label using *scutfbr*. According to the implementation in LibSVM[7], the *scutfbr* threshold tuning algorithm uses two nested 3-fold cross validation for each of the 103 labels and the classifier is trained $3 \times 3 \times 103 = 927$ times, which is infeasible in our case.

Secondly, we found that the performance of HMCN (Wehrmann et al., 2018) is sometimes much lower than expected. After tuning their model, we observed that if we first do a weighted sum of the local and global outputs and then apply the sigmoid function, HMCN's performance becomes much better (see Table 5) than doing them in the opposite order as in the paper. In addition, we found that HMCN + HAN (Yang et al., 2016) would result in extremely low performance. We had to remove HMCN's batch normalization to make it compatible with HAN. Combining HMCN with other base models did not encounter similar issues.

Table 5: Comparison of different implementations of HMCN.

| Model | RCV1 | | | Yelp | | | NYT | | |
|---|---|---|---|---|---|---|---|---|---|
| | Micro-F1 | Macro-F1 | Samples-F1 | Micro-F1 | Macro-F1 | Samples-F1 | Micro-F1 | Macro-F1 | Samples-F1 |
| HMCN (Wehrmann et al., 2018) | 78.2 | 33.2 | 78.9 | 56.3 | 8.5 | 57.3 | 62.1 | 32.4 | 62.7 |
| HMCN (Ours) | 80.8 | 54.6 | 82.2 | 66.4 | 42.7 | 67.6 | 72.2 | 47.4 | 74.2 |

Thirdly, our implementation of TextCNN (Kim, 2014) and HAN (Yang et al., 2016) shows better performance than those reported in Peng et al. (2018) due to implementation details. A comparison can be found in Table 6.

---

[7]https://www.csie.ntu.edu.tw/ cjlin/libsvmtools/multilabel/

Table 6: Comparison of different implementations of HAN and TextCNN on the RCV1 dataset.

| Model | Micro-F1 | Macro-F1 |
|---|---|---|
| TextCNN (Peng et al., 2018) | 73.2 | 39.9 |
| TextCNN (Ours) | 76.6 | 43.0 |
| HAN (Peng et al., 2018) | 69.6 | 32.7 |
| HAN (Ours) | 75.3 | 40.6 |

