# OpenReview forum: "End-to-End Hierarchical Text Classification with Label Assignment Policy"
_ICLR.cc/2019/Conference_

### Official Review · AnonReviewer2 · 2018-11-03
**reinforcement learning approach for hierarchical text classification**

**Rating:** 4
**Confidence:** 4

**Review:**

This paper presents an end to end rl approach for hierarchical text classification. The paper proposes a label assignment policy for determining the appropropriate positioning of a document in a hierarchy. It is based on capturing the global hierachical structure during training and prediction phases as against most methods which either exploit the local information or neural net approaches which ignore the hierarchical structure. It is demonstrated the method particularly works well compared to sota methods especially for macro-f1 measure which captures the label weighted performance. The approach seems original, and a detailed experimental analysis is carried out on various datasets.

Some of the concerns that I have regarding this work are :
 - The problem of hierarchical text classification is too specific, and in this regard the impact of the work seems quite limited.
 - The significance is further limited by the scale of the datasets of considered in this paper. The paper needs to evaluate against on much bigger datasets such as LSHTC datasets http://lshtc.iit.demokritos.gr/. For instance, the dataset available under LSHTC3 is in the raw format, and it would be really competitive to evaluate this method against other such as Flat SVM, and HRSVM[4] on this dataset, and those from the challenge.
- The experimental evaluation seems less convincing such as the results for HRSVM for RCV1 dataset are quite different in this paper, and that given HRSVM paper. It is 81.66/56.56 vs 72.8/38.6 reported in this paper. Given that  81.66/56.56 is not too far from that given by HiLAP, it remains a question if the extra computational complexity, and lack of scalability (?) of the proposed method is really a significant advantage over existing methods.
 - Some of the references related to taxonomy adaptation, such as [3] and reference therein,  which are also based on modifying the given taxonomy for better classification are missing.
 - Comparison with label embedding methods such as [1,2] are missing. For the scale of datasets discussed, where SVM based methods seem to be working well, it is possible that approaches [1,2] which can exploit label correlations can do even better.
[1] K. Bhatia, H. Jain, P. Kar, M. Varma, and P. Jain, Sparse Local Embeddings for Extreme Multi-label Classification, in NIPS, 2015.
[2]  H. Yu, P. Jain, P. Kar, and I. Dhillon, Large-scale Multi-label Learning with Missing Labels, in ICML, 2014.
[3] Learning Taxonomy Adaptation in Large-scale Classification, JMLR 2016.
[4] Recursive regularization for large-scale classification with hierarchical and graphical dependencies, https://dl.acm.org/citation.cfm?id=2487644

---

> ### Author Response · Authors · 2018-11-20
> **[Response to Review 2]**
>
> We very much appreciate your comments and valuable suggestions for improving the work!
>
> -------------------------------
> Q1: Regarding the problem that the scope is too narrow and the potential impact.
>
> A1: Thank you for the feedback! We would like to highlight that hierarchical text classification is an important task with a wide range of downstream applications in natural language processing and data mining (e.g., Mesh Indexing, News categorization, Law/Patent categorization)[1, 2]. In addition, the techniques developed in this work can be naturally applied to various structured prediction problems in other related domains (e.g., image categorization, user profiling, fine-grained entity typing).
>
> [1] DeepMeSH: Deep semantic representation for improving large-scale MeSH indexing Bioinformatics 2016
> [2] A Survey of Hierarchical Classification Across Different Application Domains DMKD 2011
>
> -------------------------------
> Q2: Scale of the experimental datasets.
>
> A2: Thank you for suggesting the LSHTC dataset. In terms of method scalability, HiLAP is at the same complexity as the base model (the document representation generated by the base model is reused at each step; only several extra matrix multiplications are needed) so our proposed HiLAP framework scale as long as the base model is scalable to larger datasets. We did not include comparison on LSHTC in our experiments due to two main reasons. First, we found that label hierarchy used in LSHTC is inherited from Wikipedia category system, which contains noisy information due to its crowdsourcing nature.  For example, one path in Wikipedia is “Arts -> Books -> Bookselling -> Booktown -> …”, where “Bookselling” already has nothing to do with its ancestor “Arts”. Second, authors of LSHTC modified the Wikipedia hierarchy by creating leaf node copies for many internal nodes in the original hierarchy, making it actually a “flat” label space. For example, for one page named “Book tour” originally at internal node “Bookselling”, a pseudo-node “Bookselling*” would be added under “Bookselling”, and “Book tour” will be moved to the leaf “Bookselling*”.
>
> -------------------------------
> Q3: Regarding experiment results of HRSVM on the RCV1 dataset.
>
> A3: Thank you for bringing out this question and sorry about the confusion here. The RCV1 dataset used in HRSVM work [1] is not the same version as the one that was used in our work and most existing work. The numbers reported in [1] are thus not comparable. The 103 labels in the original RCV1 dataset were extended to 137 labels in [1]. Experiments reported in [2] show that same model can obtain up to 23 Macro F1 improvement on the 137-label version, compared to the 103-label version. We have clarified the details about the dataset in our updated version.
>
> [1]Recursive Regularization for Large-scale Classification with Hierarchical and Graphical Dependencies KDD 2013
> [2]Large-Scale Hierarchical Text Classification with Deep Graph-CNN WWW 2018
>
> -------------------------------
> Q4: Some of the references related to taxonomy adaptation, such as [3] and reference therein,  which are also based on modifying the given taxonomy for better classification are missing.
>
> A4: Thanks for pointing out these relevant work. The line of research on taxonomy adaptation is related but has a different goal as compared to our problem setting---they aim to modify the given label hierarchy (by pruning nodes in the tree) to output a new label hierarchy which is better suited for classification. Our work, however, deals with a fixed label hierarchy and focuses on designing a label assignment policy to traverse it in a smart way to predict labels for a document. We updated the paper to clarify this and added these references to discuss the commonalities and differences.
>
> -------------------------------
> Q5: Comparison with label embedding methods such as [1,2] are missing. For the scale of datasets discussed, where SVM based methods seem to be working well, it is possible that approaches [1,2] which can exploit label correlations can do even better.
>
> A5: Thank you for pointing out this line of work. However, these work focus on the scenario where there are a large number of “independent” labels (i.e., a flat label space). They do not leverage label hierarchy or any information about the structured dependencies between the labels, which is the focus of our work.  For example, [2] assumes that such a label hierarchy is not available in their setup. We have included them in our related work discussion and clarified the distinctions.

---

### Official Review · AnonReviewer3 · 2018-11-04
**Interesting idea, but would like to see clearer set of claims with appropriate evaluation.**

**Rating:** 4
**Confidence:** 5

**Review:**

This work proposes an RL approach for hierarchical text classification by learning to navigating the hierarchy given a document. Experiments on 3 datasets show better performance. I'm happy to see that it was possible to

1. "we optimize the holistic metrics over the hierarchy by providing the policy network with holistic rewards"

I don't quite understand what are the "holistic metrics" and "holistic rewards". I would like the authors to answer "what exactly does reinforcement learning get us ?"
 - Is it optimizing F1 metric or is it the ability to fix inconsistent labeling problem ?
- If it is the latter, what is an example of inconsistent labeling, what fraction of errors (in table 2/3) are inconsistent errors. Are we really seeing the inconsistent errors drop ?
- If it is the former, how does this compare to existing approaches for optimizing F1 metric.

2. "the F1 score of each sample xi"

a. F1 is a population metric, what does it mean to have F1 for a single sample ?
b. I'm not aware of any work that shows optimizing per-example f_1 minimizes f_1 metric over a sample.

3. with 10 roll-outs per training sample, imho, it seems unrealistic that the expected reward can be computed correctly. Would'nt most of the reward just be zero ? Or is it the case the model is initialized with an MLE pretrained parameters (which seems like it, but im not too sure).

Results analysis,
- imho, most of the rows in Table 2 does not seem comparable with each other due to pretrained word-embeddings and dataset filtering, e.g. SVM-variants, HLSTM.
- in addition to above, there is the standard issue of using different #parameters across models which increases/decreases model capacity. This is ok as long as all parameters were tuned on held out set, or using a common well established unfiltered test set - neither of which is clear to me.
- it is not clear how the F1 metric captures inconsistent labeling, which seems to be the main selling point for hi-lap.

side comment
- reg textcnn performance, could it be that dropout is too high ? (the code was set to 0.5)

---

> ### Author Response · Authors · 2018-11-20
> **[Response to Review 3] - Method**
>
> We really appreciate your detailed comments and valuable feedback!
>
> -------------------------------
> Q1: Regarding “what reinforcement learning gets us.”
>
>
> A1: Existing works largely maximize the accuracy of labels and have a gap between training/test (e.g., top-down approaches train a set of classifiers independently and only consider the label dependencies during inference). In our method, training and inference are consistent and label dependencies are captured at both phases. We use RL as a tool to achieve such goals by designing a label assignment policy and rewarding the agent with the per-sample f1 score (please see definition in A3). We show that the performance can be improved through the exploration of the label hierarchy and model of label dependencies. In particular, reward shaping allows us to emphasize the importance of coarse-grained labels (since the label assignment begins at the root node) while prior works treat each label equally (by measuring accuracy).
>
> “providing the policy network with holistic rewards” indicates the former, i.e., we can explicitly optimize the per-sample f1 score, which reflects the overall quality of the label assignment of one sample. As far as we know, there are no existing approaches that explicitly optimize F1 metric for hierarchical classification.
>
>
> -------------------------------
> Q2: Regarding fixing inconsistent labeling issue.
>
> A2: Sorry for the confusion. The ability to fix the inconsistent labeling problem is inherently built in the label assignment policy. By following the policy (Figure 1 and Sec 2.1), there will never be inconsistent labels. In fact, the label inconsistency is more of a practical issue. In reality, one doesn’t want to tell the users that an instance is an apple but in the meantime, not a fruit. In terms of performance, correcting such issues naively doesn’t always provide much performance gain. E.g., if we simply add all the ancestors of predicted nodes (if these ancestors haven’t been predicted) or remove those nodes (if one of their ancestors hasn’t been predicted), there would be about ~1 F1 change (on a 100 scale). And such correction could either lead to performance gain or drop. On the contrary, we solved this problem from the root.
>
> -------------------------------
> Q3. Regarding the definition of “sample F1” and the optimization of sample F1.
>
> A3. Sorry about the confusion caused by the “sample F1” metric. The sample F1 is defined for each instance in multi-label classification setting. For example, if the gold labels of instance xi are (1,2,3) and the predicted labels are (2, 3, 4, 5), its precision and recall would be 2/4 and 2/3, respectively. We adopted this name based on its use in sklearn [1]. The same metric is referred to as EBF in [2,3], and example-based F1 in LSHTC [4]. The RL reward is designed in a way that aims to capture the sample F1 metric.  We have updated the description of this evaluation metric to make it more clear.
>
> [1] http://scikit-learn.org/stable/modules/generated/sklearn.metrics.f1_score.html
> [2] MeSHLabeler: Improving the accuracy of large-scale MeSH indexing by integrating diverse evidence Bioinformatics 2015
> [3] DeepMeSH: Deep semantic representation for improving large-scale MeSH indexing Bioinformatics 2016
> [4] LSHTC: A Benchmark for Large-Scale Text Classification CoRR 2015

---

> > ### Comment · AnonReviewer3 · 2018-11-29
> > **Clarifying my questions since I dont feel like it was answered.**
> >
> > Q1/Q2: Going back to my questions,
> > "Is it optimizing F1 metric or is it the ability to fix inconsistent labeling problem ? "
> > Based on the authors response it looks like the answer is both. If yes, it is natural to tease out the effect of each, for e.g. without the hierarchy part, how much is rl helping improve per example f1.
> >
> > "If it is the latter, what is an example of inconsistent labeling, what fraction of errors (in table 2/3) are inconsistent errors. Are we really seeing the inconsistent errors drop ?"
> > I did not get the author's response to what fraction of errors in other methods are due to inconsistent labeling.
> >
> > - If it is the former, how does this compare to existing approaches for optimizing F1 metric.
> > I think this question still remains.
> >
> > Q3: Thanks for clarifying !

---

> > > ### Author Response · Authors · 2018-12-03
> > > **We very much appreciate you for pointing out this unclear answer.**
> > >
> > > We very much appreciate you for pointing out this unclear answer. To further clarify our responses:
> > >
> > > First, “label consistency” is guaranteed by our proposed label assignment policy, e.g., if a label is assigned to the document then its ancestor label must be also assigned. We apply RL to learn such a policy network to achieve the goal, as it cannot be learned using prior supervised learning objectives. In particular, to answer your question on “how much gain can we get by fixing label inconsistency issue”, we further conducted experiments to check the percentage of test documents that are predicted with inconsistent labels. For example, we found 29186/781265 (3.7%) predictions have inconsistent labels for TextCNN on RCV1. In contrast, our method HiLAP always ensures 0% label inconsistency. We are aware that such error rate does not reflect the F1 metrics, but it can provide a rough picture of how severe the issue is in existing methods. More details of label inconsistencies and ways of correcting them can be found in [1].
> > >
> > > Second, sample F1 is a non-differentiable metric and thus we apply RL to optimize it. Loss function in some prior work can be seen as optimizing the multi-label classification accuracy (or “subset 0/1 accuracy”) [2], and is thus relatively more sensitive to label bias. To the best of our knowledge, we are the first to propose optimizing F1 metric for hierarchical classification.
> > >
> > > [1] A Survey of Hierarchical Classification Across Different Application Domains
> > > [2] SeCSeq: Semantic Coding for Sequence-to-Sequence based Extreme Multi-label Classification

---

> ### Author Response · Authors · 2018-11-20
> **[Response to Review 3] - Experiment**
>
> -------------------------------
> Q4: “With 10 roll-outs per training sample, imho, it seems unrealistic that the expected reward can be computed correctly. Would'nt most of the reward just be zero ? Or is it the case the model is initialized with an MLE pretrained parameters (which seems like it, but im not too sure).”
>
> A4: Yes, the model is initialized with pre-trained parameters. The 10 roll-outs are independently performed and should be similar to each other with slight variance (when the distribution is pretrained). The same approach is used in [5,6].
>
> [5] End-to-End Reinforcement Learning for Automatic Taxonomy Induction ACL 2018
> [6] Go for a Walk and Arrive at the Answer: Reasoning Over Paths in Knowledge Bases using Reinforcement Learning ICLR 2018
>
>
> -------------------------------
> Q5: “..., most of the rows in Table 2 does not seem comparable with each other due to pretrained word-embeddings and dataset filtering, e.g. SVM-variants, HLSTM.”
>
> A5: There is no data filtering for RCV1. We follow the original training/test split, which is commonly used in most previous work, including those reported by [9]. The dimension sizes of word-embeddings are both 50 in our paper and [9]. In fact, [9] retrained the word-embeddings on the stemmed corpus, which they claimed to be better than original words and we reported their performance on the stemmed version. In that sense, their performance is expected to be even lower under the same setup.
>
> -------------------------------
> Q6:  “in addition to above, there is the standard issue of using different #parameters across models which increases/decreases model capacity. This is ok as long as all parameters were tuned on held out set, or using a common well established unfiltered test set - neither of which is clear to me.”
>
> A6: Sorry about the unclear description. Since there is no well-established validation set, we randomly sample a portion of the training set as the held-out set, as also adopted by prior work [7,8,9]. In particular, HiLAP and the base model use exactly the same hyperparameters all the time. We have updated the paper to include these details.
>
> [7] Semi-supervised Convolutional Neural Networks for Text Categorization via Region Embedding NIPS 2015
> [8] Effective Use of Word Order for Text Categorization with Convolutional Neural Networks NAACL 2015
> [9] Large-Scale Hierarchical Text Classification with Deep Graph-CNN WWW 2018
>
> -------------------------------
> Q7: “It is not clear how the F1 metric captures inconsistent labeling, which seems to be the main selling point for hi-lap.”
>
> A7: Please refer to our answer in A2.
>
>
> -------------------------------
> Q8 : “regarding text-CNN performance, could it be that dropout is too high ? (the code was set to 0.5)”
>
> A8: We tried reducing the dropout but did not observe clear performance changes. We note that [10] also sets the dropout to 0.5. That being said, HiLAP is built on top of the base model and reuse all the parameters of the base model. HiLAP (textcnn), for example, uses the same dropout rate (0.5) and should be lower/higher in performance as well.
>
> [10] Deep Learning for Extreme Multi-label Text Classification SIGIR 2017

---

> > ### Comment · AnonReviewer3 · 2018-11-29
> > **thanks for clarifying, one q still remains**
> >
> > Q5/Q6:
> >
> > "most of the rows in Table 2 does not seem comparable with each other "
> >
> > I am not able to find a good answer to this question. Different models seems to have different embedding size and #parameters and hence different model capacity. This will most definitely cause different performances on the test set.
> >
> > One way to fix this issue - which is not applicable here, nor in any work mentioned in Table 2, is to use the exact same identical test across all works; for e.g. most of the works results on imagenet are comparable with each other despite using different parameters since they use an identical test set.

---

> > > ### Author Response · Authors · 2018-12-03
> > > **Thank you very much for the reply!**
> > >
> > > Thank you very much for the reply!
> > >
> > > First, we want to clarify that the test set used in all experiments reported in Table 2 is *identical*--we adopted the test set given by the RCV1 dataset and so did the other compared methods in Table 2.
> > >
> > > Second, we understand that methods compared in Table 2 have different model capacity (due to differences in model architecture, etc.). However, Table 2 aims to present our validation and justification on two main things: (1) our proposed framework can build on top of existing base models (e.g., TextCNN, HAN and bow-CNN) to improve *in the setting of hierarchical classification* (thus, we show different variants of our method using different base models). These comparisons, in particular, mitigate the differences in model capacity; and (2) our best-performing method, HiLAP (bow-CNN), can advance the state-of-the-art method reported on RCV1 (thus, the set of various methods we compared).
> > >
> > > We will emphasize this detail in our final version, and really hope you could consider our clarification on this problem.

---

### Official Review · AnonReviewer1 · 2018-11-06
**Clever and promising techniques to force the inference process in structured classification to converge, but experiments seem to lack apple-to-apple comparisons**

**Rating:** 5
**Confidence:** 4

**Review:**

This papers uses the label hierarchy to drive the search process over a set of labels using reinforcement learning. The approach offers clever and promising techniques to force the inference process in structured classification to converge, but experiments seem to lack apple-to-apple comparisons.

However, I think the authors should rather present this work as structured classification, as labels dependencies not modeled by the hierarchy are exploited, and as other graph structure could be exploited to drive the RL search.
I tend to see hierarchical classification as an approach to multi-label classification justified by a greedy decomposition that reduced both training and test time. This view has been outmoded for more than an decade, first as flat approaches became feasible, and now as end-to-end  structured classification is implementable with DNNs (see for instance David Belanger work with McCallum)

Compared to other structured classification approaches whose scope is limited by the complexity of the inference process, this approaches is very attractive. The authors open the optimization black box of the inference process by adding a few very clever tricks that facilitate convergence:
- Intermediate rewards based on the gain on F1 score
- Self critical training approach
- "Clamped" pre-training enabled by the use of state embeddings that are multiplied my a transition to any state in the free mode, and just the next states in the hierarchy in the clamped mode
- Addition of a flat loss to improve the quality of the document representation

While those tricks may have been used for other applications, they seem new in the context of hierarchical/multi-label/structured classification.

While the experiments appear thorough, they could be the major weakness of this paper. The results the authors quote as representative of other approaches seem in fact entirely reproduced on datasets that were not used on the original papers, and the authors do not try an apple-to-apple comparison to determine if this 'reproduction' is fair. None of the quoted work used the 2018 version of Yelp, and I could only find RCV1 Micro-F1 experiments in Johnson and Yang, who report a 84% micro-F1, far better than the 76.6% reported on their behalf here, and better than the 82.7% reported  by the authors. I read note 4 about the difference in the way the threshold is computed, but I doubt it can explain such a large difference. I did not check everything, but could not find and apple-to-apple comparison?

Have the network architecture been properly optimized in terms of hyper-parameters?
In particular, having tried Kim CNN on large label sets, I suspect the author settings using a single layer after the convolution is sub-optimal. I concur with the following paper than an additional hidden layer is essential: Liu et al "Deep Learning for Extreme Multi-label Text Classification". I also note the 32 batch size could be way too small for sparse label sets (I tend to use a batch size of 512 on this type of data).

---

> ### Author Response · Authors · 2018-11-20
> **[Response to Reviewer 1]**
>
> Thank you for your suggestions on improving the presentation of our work! We would like to conduct experiments on other structured prediction tasks in the future using the same philosophy.
>
> -------------------------------
> Q1: About experiment datasets and our reported performance for baseline methods.
>
> A1: RCV1 is one of the few well-used datasets for hierarchical classification and we followed the original training/test split. We are not sure what 76.6% is referred to in “far better than the 76.6%”. Threshold tuning does affect the performance but is time-consuming in the meantime. [1] avoids this issue (but didn’t solve it) by evaluating AUC instead. We couldn’t reproduce an 84% micro-F1 of bow-CNN without threshold tuning and the best we could get is 82.7%. Based on the 82.7% base model, HiLAP (bow-cnn) then improves its performance to 83.3%.
>
> The main aim of comparing HiLAP with the base models is to show that we can improve upon them by exploring the label hierarchy. For example, equipping HAN with HiLAP achieves similar performance to HR-DGCNN [2] (which also models the hierarchy) even though the original HAN is worse than HR-DGCNN. Similarly, for other base models, we can constantly improve their performance although the architecture and hyper-parameters for document representation are unchanged.
>
> The “apple-to-apple” comparison is mainly between HMCN [1], HR-DGCNN[2] (the most recent state-of-the-art methods on hierarchical classification) and HiLAP. We updated the description of baselines to make it more clear.
>
> [1] Hierarchical Multi-Label Classification Networks ICML 2018
> [2] Large-Scale Hierarchical Text Classification with Deep Graph-CNN WWW 2018
>
> -------------------------------
> Q2: Have the network architecture been properly optimized in terms of hyper-parameters?
>
> A2: Thank you very much for your valuable suggestions. We tuned the hyper-parameters on a held-out development set (such as learning rate, regularization, and vector dimensions).  We believe the comparison between HiLAP and corresponding base model is relatively fair because they use exactly the same hyper-parameters throughout the experiment comparison.  If we add an additional hidden layer to Kim CNN (which more or less turns it into XML-CNN[3] already, except for the dynamic pooling), the same hidden layer would also be added to HiLAP (Kim CNN). The same rule applies to the batch size. We leave such changes as further exploration.
>
> Performance change w.r.t learning rate and regularization
> -------------------------------------------------------------------------------------------------------------
> learning rate	        | 1e-3	| 5e-4	| 1e-4	| 2e-3	| 1e-3	| 1e-3	 | 1e-3	|
> weight_decay	| 1e-6	| 1e-6	| 1e-6	| 1e-6	| 5e-6	| 5e-7	 | 1e-7	|
> Micro-F1		| 82.73	| 82.51	| 80.86	| 82.44	| 82.48	| 82.6	 | 82.2	|
>
>
>
> [3]Deep Learning for Extreme Multi-label Text Classification SIGIR 2017

---

### Author Response · Authors · 2018-10-15
**Code released**

We released our code in an anonymized repo: https://github.com/hi-label-assignment-policy/HiLAP

---

### Meta-Review · Area_Chair1 · 2018-12-14
**author response did not address the major concerns of the reviewers regarding the empirical results**

**Confidence:** 5
**Recommendation:** Reject

**Metareview:**

This paper presents a reinforcement learning approach to hierarchical text classification.

Pros: A potentially interesting idea to drive the search process over a hierachical set of labels using reinforcement learning.

Cons: The major concensus among all reviewers was that there were various concerns about experimental results, e.g., apple-to-apple comparisons against prior art (R1), proper tuning of hyper-parameters (R1, R2), the label space is too small (539) to have practical significance compared to tens of thousands of labels that have been used in other related work (R3), and other missing baselines (R3). In addition, even after the rebuttal, some of the technical clarity issues have not been fully resolved, e.g., what the proposed method is actually doing (optimizing F1 metric vs the ability to fix inconsistent labeling problem).

Verdict:
Reject. While authors came back with many detailed responses, they were not enough to address the major concerns reviewers had about the empirical significance of this work.